# Performing Borders: Queer and Trans Experiences at the Canadian Border

**Edwin Hodge** [1,*]**, Helga Hallgrimsdottir** [2] **and Marianne Much** [3]

1   Department of Sociology, University of Victoria, Victoria, BC V8W 3P5, Canada
2   School of Public Administration, University of Victoria, Victoria, BC V8W 2Y2, Canada
3   Centre for Global Studies, University of Victoria, Victoria, BC V8W 3P5, Canada
*   Correspondence: edhodge@uvic.ca

**Abstract:** Biometric security and screening systems have revolutionized border crossings. As bodies move across the physical space of the borderland, the border moves through them, scanning and cataloguing and scrutinizing bodies for irregularity. While such technologies have been scrutinized, they have largely been so through heteronormative and cisnormative lenses that fail to recognize the vastly different experiences of nonbinary, nonconforming, transgender, and queer border crossers. This paper examines the implications of what we argue is the individualization of the border, and the effects of biometric security screenings for people whose bodies do not conform to heteronormative and cisnormative standards. We argue that border securitization increasingly equates body differences to narratives of threat and risk, which endangers nonbinary, trans, and queer border crossers, and places their safe passage at risk.

**Keywords:** biometric borders; transgender; gender-nonconforming; nonbinary; queer; securitization; risk

---

## 1. Introduction

Borders represent perhaps one of the greatest paradoxes of our time: While barriers to economic flows are dismantled, it is clear that borders remain highly significant as arbiters of human flows. Recent events, including the Schengen crisis in Europe, as well as the "build the wall" campaign in the United States, reveal that borders have a persistent role as pillars of national identity, sites of exclusion and expulsion, and as mechanisms to create spaces of exception.[1] The border represents a physical, spatial, and symbolic marker between those who are in, and who are not and provides a focal point for concerns for the policing and securing of these boundaries.[2] Border processes—especially as they pertain to border-crossers and migrants—are thus far from neutral arbiters of flows. Following Hannah Arendt's observation that sovereignty is "nowhere more absolute" than in the processes through which states expel and deny access to their territory, the border is a key mechanism for stratifying human mobility and in so clarifies and articulates the rights and claims of those who are already inside a territory.[3] Borders thus both generate and bound citizenship.

Yet, as Salter has argued, the border represents a perpetual 'state of exception': At the moment at which we cross a border, the legal norms and rights of citizenship are suspended.[4] This extra-legality of the border-zone is heightened and intensified for border-crossers who have precarious and contingent

---

1   (Agamben 1998; Parker and Vaughan-Williams 2009).
2   (Anderson et al. 2011; Pickering and Weber 2006).
3   (Arendt 1973).
4   (Salter 2008; Basaran 2008).

claims to citizenship, especially refugees. The purpose of this paper is to examine how the border operates as an arbiter of the rights of refugees claiming asylum on the basis of sexual orientation, gender identity or expression (SOGIE).

Such claims are made even more difficult when cultural factors are added to the mix; in many countries where sexual orientation or gender identity are sources of discrimination or oppression, they are also the focus of cultural taboos, making public acknowledgment a source of intense shame. To compound this already thorny issue, immigration and refugee agents in Canada have, for many years, operated under a legal regime formed of a patchwork of legal statutes, court rulings, departmental guidelines, and memoranda. As a result, SOGIE claimants may be faced with rejection of their applications due to a lack of clearly established procedural guidelines, rather than from a lack of demonstrated need or justified fear of persecution. This risk is heightened for claimants whose gender identity places them outside the narrow confines of Western patterns of heteronormativity—whose queerness (and very reason for their attempts to claim refugee status) establishes their bodies and identities as different and therefore subject to greater levels of scrutiny.

It is this challenge to the Canadian immigration and refugee system that impelled the investigation presented in this paper: the examination of the conditions and challenges faced by queer asylum seekers as they attempt to navigate their crossing of the Canadian border. Through this investigation, we illustrate the dynamic relationship between borders and bodies—specifically the bodies of queer and transgender refugee claimants—that illustrates the ways in which borders and the practice of navigating borders is performative, as gender itself is performative. Further, we illustrate how these performative interactions extend beyond the immediate intersection of queer and transgender bodies with the geopolitical borderline and the agents responsible for policing it, drawing out the performance of bordering through time and space such that the borderland of queer and transgender crossings becomes a four-dimensional maze wherein claimants must at different times contend with competing fragments of legislation or bureaucratic 'guidelines'.

These challenges have not gone unrecognized by the Canadian government. In early 2017, the Immigration and Refugee Board of Canada (IRB) released an updated set of guidelines and procedures for its personnel and included in this updated guide was a section entitled "Proceedings before the IRB involving Sexual Orientation and Gender Identity and Expression". The purpose of the guidelines was, in the words of the IRB:

> ... to promote greater understanding of cases involving sexual orientation and gender identity and expression (SOGIE) and the harm individuals may face in presenting their cases before the Immigration and Refugee Board of Canada (IRB) and establishes guiding principles for decision-makers in adjudicating cases involving SOGIE.[5]

The details of these new guidelines provide an important glimpse into the operational realities of the IRB and are at once a validation of the unique experiences of trans, queer, and nonbinary refugee claimants and an area for concern for that very group, due to an apparent contradiction between the wording of these guidelines and an earlier precedent in Canadian law.

In 1993, the Supreme Court of Canada ruled in *AG v. Ward* that a "particular social group" (a term sitting at the heart of Canadian refugee law) should be interpreted to mean " ... groups defined by an *innate* or *unchangeable* characteristic" (emphasis ours).[6] For queer and transgender refugee applicants, their status as belonging to "a particular social group", while prima facie true, is problematic considering the legal meaning of that term. In effect, the legal precedent laid down in Ward establishes that refugee claimants attempting to use their gender or sexual orientation as grounds for seeking asylum can do so only if they are able to convince individual refugee claims officers that their gender

---

[5]  (Immigration and Refugee Board of Canada 2017).
[6]  (Attorney General v. Ward 1993).

or sexual orientation are 'innate' or 'unchangeable', a position that many transgender claimants are unable or unwilling to adopt.

Effectively, the Canadian refugee system's current practice erases transgender and gender-nonconforming claimants' identities by relying on outdated and incorrect assessments of what their identities 'ought' to look like under Canadian law. Where queer, trans, nonbinary, and nonconforming gender identities are rooted in fluidity, Canadian refugee protections for gender and sexual minorities requires such identifications to be fixed and immutable—two criteria that omit queer claimants' identities. In effect, this results in claimants whose identities are fluid being forced to conform to rigid patterns of gender identity and expression imposed upon them at the border. Further, while subsequent attempts at correcting this glaring legislative gap in refugee law do provide officials with greater direction in the processing of SOGIE claims, the resultant legislative landscape remains fractured and unfair to claimants whose identities—and therefore the foundations for their claims of persecution—remain ill-defined by the very systems on which their petitions for asylum rest.

We argue that the impact that Canada's piecemeal and incomplete approach to refugee law and the asylum-seeking process has on queer and transgender claimants is that their identities are effectively rendered invisible by a system that lacks sufficient understanding of the realities of transgender claimants' experiences and identities. While queer, trans, and nonbinary, nonconforming claimants' gender identities and expression—and frequently sexual identities and expression—emerge from socially constructed narratives of sex and gender, the Canadian refugee system relies on categorizations of these identities grounded in understandings that are both deterministic and essentialist. We argue that the ad hoc nature of contemporary refugee and asylum policy in Canada is, with respect to queer and transgender claimants—and indeed most other gender-nonconforming identities—critically underdeveloped.

## 2. History

The story of immigration in Canada is replete with examples of exclusionary measures directed against marginalized populations.[7] The issue for queer and trans populations, however, is even more complicated, as the cultural and political recognition of these populations is relatively recent. While there is a long history of gender variance globally, the increased rationalization and bureaucratization of nation-building projects, coupled with the medicalization—and pathologization—of gender and sexuality in the late 19th and 20th centuries across both Europe itself and where Europeans had colonized added a new dimension to the process of gender variant bodies crossing national borders.

In the Canadian context, the specific issue—and problematizing—of queer and trans immigration emerged during the Cold War, when the Canadian government, along with Royal Canadian Mounted Police (RCMP) border agents, became concerned with screening out applicants who were believed to represent a specific security risk: "character weakness".[8] In December, 1950, a government committee in charge of revising the Immigration Act added a new prohibited class: "homosexuals, lesbians, and persons coming to Canada for any immoral purpose".[9] The version assented on 4 July 1952 had the following clauses for prohibited persons in Section 5(e): "prostitutes, homosexuals, or persons living on the avails of prostitution or homosexualism (sic), pimps, or persons coming to Canada for these or any other immoral purposes . . . "[10]

The issue of queer and trans refugee asylum was not a concern for Western states until the United Nations Higher Commissioner for Refugees (UNHCR) convention was ratified by 1954. The introductory note to the UNHCR convention states that:

---

7　(Luibheid 2005).
8　(Robinson and Kimmel 1994).
9　(Robinson and Kimmel 1994).
10　(Canadian Museum of Immigration at Pier 21 2017.)

. . . it was a post-Second World War instrument" limited "in scope to persons fleeing events occurring before 1 January 1951 and within Europe. The 1967 Protocol removed these limitations and thus gave the Conventions universal coverage.[11]

The convention defined a refugee as someone "who is unable or unwilling to return to their country of origin owing to a well-founded fear of being persecuted for reasons of race, religion, nationality, membership to a particular social group, or political opinion";[12] the commitment to the protection of "particular social groups" was added at the urging of Sweden, whose representative stated that " . . . experience has shown that certain refugees had been persecuted because they belonged to particular social groups . . . Such cases existed, and it would be as well to mention them explicitly"[13]. This updated definition was important, as it assisted diplomats, advocates, and other stakeholders in creating a definition of 'refugee' that was not tied solely to nationality and, when applied, expanded refugee status to those who have a basis to fear persecution, rather than tying refugee status to a specific crisis or nationality, and expanded the list of reasons that would warrant the application of refugee status.[14]

In effect, the changes to the convention meant that members of groups falling outside of the categories of race, religion, political opinion or nationality had become eligible to claim refugee status, dramatically expanding the scope of what sorts of categories of people could do so. As with other international agreements, the implementation of the convention—as well as its specific interpretations vis a vis a state's obligations—fell to individual members to implement according to their own legal traditions. In Canada, refugee claims making use of the "membership to a specific social group" provision tend to be informed by two principle texts: the *Matter of Acosta* decision and James Hathaway's interpretative text, *The Law of Refugee Status.*

The 1985 case *Matter of Acosta* was cited as key precedent by the Canadian scholar James Hathaway and then by *AG v. Ward*, which cited both *Acosta* and Hathaway. *Acosta* was different in its approach to defining a particular social group because it applied the principle of *eiusdem generis*—or "of the same kind or nature" —to its interpretation of the nondiscrimination principle. In other words, to be a member of a "particular social group" was to be treated no differently than a claim rooted in a person's race, religion or nationality. Since these latter categories were treated as 'unchangeable' or 'immutable' characteristics of a person's identity, *eiusdem generis* had the practical effect of making membership in particular social groups 'immutable' as well.

In effect, this ruling meant that "membership to a specific social group" required that membership be tied to characteristics that were understood to be immutable or fundamental.[15] In applying principles of nondiscrimination and weighing immutable or fundamental characteristics, membership in groups not listed in the convention can meet the definition of a refugee, namely, claims based on gender or sexuality. In the *Matter of Acosta*, the US Department of Justice deportation board, in their decision wrote that:

> The other grounds of persecution in the Act and the Protocol listed in association with 'membership in a particular social group' are persecution on account of 'race', 'religion', 'nationality', and 'political opinion'. Each of these grounds describes persecution aimed an immutable characteristic: a characteristic that either is beyond the power of an individual to change or is so fundamental to individual identity or conscience that it ought not be required to change.[16]

---

[11] (United Nations High Commissioner for Refugees UNHCR).
[12] (United Nations High Commissioner for Refugees UNHCR).
[13] (Hathaway and Foster 1991).
[14] (Fullerton 1993).
[15] (Musalo 2003).
[16] (United States Department of Justice 1985.)

*Acosta* thus established a legal precedent that linked gender and sexual identity—in all its forms and variations—to immutable, essentialist articulations of gender and sexual identity. Under this legal regime, claimants can only 'rightly' be considered refugees if their fears of persecution can be shown to be related to an *immutable* characteristic of their being.

The development of jurisprudence which opened the way for refugee claims based on sexual orientation thus took place in an extremely narrow and limited context. Interpretations like *Acosta* emerged from a period where the legal exclusion of homosexuals was still in effect throughout the Global North. In the United States, for example, it was not until the 1990 Immigration Act that gays and lesbians were no longer excluded from migration; it was not until 2010 that HIV-positive people were no longer an excluded class. While still denying the rights of homosexual citizens, Canada, in 1976, amended the Immigration Act by adopting the definition of a refugee as defined by the UN Convention Relating to the Status of Refugees and removing homosexuals as an excluded class.

Yet despite these alterations (and modernizations) to the Immigration Act, the experiences of gender-nonconforming and trans refugees at the borders remain fraught with difficulty; though jurisprudential shifts have been observed, they have often trailed cultural shifts and developments and as a result have often lagged far behind more recent changes in the understanding of sexuality, sexual orientation, and gender identity. Beyond the slow pace of legislative change, cultural attitudes regarding queer, trans, and gender-nonconforming people's identities and experiences often remain negative, adding an additional layer of difficulty to an already marginalized group of people as they seek to navigate the Canadian refugee system.

## 3. Biometrics, 'Suspicious' Bodies, and the Construction of the 'Ideal' Citizen

Since the events of 9/11, border securitization has become the standard—indeed the principle—region of concern for North American states. This preoccupation was clearly spelled out in agreements such as the Smart Border Accord of 2001, which emphasized the role of the border as a security barrier, even as it attempted to maintain pre-9/11 levels of cross-border trade.[17] While much of the language of such agreements emphasizes the permanence—and selective permeability—of the physical border (through checkpoints, customs agents and other physical and institutional barriers at points of entry), many of the initiatives had the effect of 'stretching' the border deep inside the state, through programs such as the NEXUS and FAST preclearance programs.[18] Like border checkpoints and other physical barriers to entry, preclearance programs form another layer of a nation's security regime, but unlike physical border controls, preclearance systems in effect distantiate the individual from the category of claims-seeker. As with "intent-management" at the Australian border (where pre-clearance type strategies are used to police and stratify entry into Australia prior to the border crossing), pre-clearance systems prioritize border-crossers demographic characteristics (i.e., membership in suspect groups) over individual claims.[19]

The irony that presents itself here is that this shift in regulatory scrutiny focuses attention on *bodies* at the most granular level possible: the level of the individual. Preclearance regimes often include certain body-specific preconditions for would-be 'safe travelers'[20], including specific wording aimed at establishing visual correspondence between a person's physical appearance, and the (presumed immutable) identifying characteristics on government issued identification. Consider the language of Transport Canada's 2011 amendment to its Identity Screening Regulations, which state:

---

17　(Salter and Piche 2011).
18　(Salter and Piche 2011).
19　(Pickering and Weber 2014).
20　(Salter and Piche 2011).

> A) An air carrier shall not transport a passenger if ... the passenger does not appear to be of the gender indicated on the identification he or she presents (Verification of identity, section 7(1))
>
> B) ... an air carrier may transport a passenger who presents a piece of photo identification but does not resemble the photograph if ... the passenger's appearance changed for medical reasons after the photograph was taken and the passenger presents the air carrier with a document signed by a health care professional and attesting to that fact.[21]

Though this language was repealed in subsequent amendments to the law, its existence illustrated the extent to which securitization had become inextricable from issues of gender, not to mention race, class, sexual orientation, and ability.[22] As critics of the language of the law argued, preclearance stipulations, not to mention at-the-gate security measures designed to screen for suspicious activity, had the effect of singling out pre- and non-operative trans people, as well as gender-nonconforming and queer travelers as suspicious *by definition*.[23]

Despite the repeal of this language, the gendered nature of border securitization remains unchanged. In the language of the updated Immigration and Refugee Board of Canada Chairperson's Guidelines, a similar, problematic language exists,

> ... to promote greater understanding of cases involving sexual orientation and gender identity and expression (SOGIE) and the harm individuals may face in presenting their cases before the Immigration and Refugee Board of Canada (IRB) and establishes guiding principles for decision-makers in adjudicating cases involving SOGIE.

That the IRB recognized an area of concern is laudable, yet the attempts at addressing it have, paradoxically, reified the problem by entrenching gender as a space where difference becomes inextricably bound to risk. In other words, the rise of the 'biometric border' as a new frontier in border securitization has resulted in a reality where bodies themselves become sites that enclose multiple boundaries.[24] The result is that the biometric border:

> ... is the portable border par excellence, carried by mobile bodies at the very time as it is deployed to divide bodies at international boundaries, airports, railway stations, on subways or city streets, in the office or the neighborhood.[25]

The impacts of bordering processes on bodies are significant, especially for marginalized, non-normative, and nonconforming bodies. Those travelers or refugees whose bodies challenge normative conventions of sex and gender, such as a person who was assigned male at birth but presenting as feminine or read as "woman", may trigger a security response in spaces that have evolved almost exclusively to scrutinize bodies for 'irregularities'.[26] Since security biometrics rely on standardized patterns of bodies, deviations constitute a de facto rational for additional scrutiny and explanation. Nonconforming bodies become subjects of pathologization and concern by virtue of their existence. The apparatus of biometric bordering creates a contentious space where some bodies can be declared "out of place" and therefore subject to increased scrutiny or even a refusal to grant entry at all, as such bodies represent "anomalies" that trigger a host of security protocols in ports of entry such as airports.[27] Yet the categorization of variant bodies as "anomalous" under biometric protocols carries a deeper philosophical concern: If biometrics are increasingly used to screen "anomalous" bodies, then

---

[21] (Government of Canada 2017).
[22] (Baker 2013).
[23] (Raj 2012).
[24] (Amoore 2006).
[25] (Amoore 2006).
[26] (Beauchamp 2016).
[27] (Currah and Mulqueen 2011).

the barriers experienced by trans and nonconforming travelers are unlikely to remain isolated to them for long. Further, what are the implications of screening variant bodies when it comes to admitting and excluding people from avenues to citizenship? What does the "ideal" Canadian immigrant look like?

Biometric bordering has two objectives: The first, as scholar Holger Pötzsch argues, is to "*verify* identities by comparing passport or ID card information with the biometric characteristics with the individual carrying the documents".[28] The second purpose of the biometric border is to *establish* the reality of identity through the collection of personal information of all kinds to construct composite images of a person's day-to-day interactions.[29] Relying on the growing body of electronic data about each individual, these technologies may potentially turn each body into "data-doubles, the enactments of which entail real material effects".[30] Such effects manifest into the question of border mobility, which can differ from one identity to another. Everyone, depending on race, gender, sexuality or class experiences the border and mobility differently.

In addition to profiling based on intersections of race, gender, sexual orientation, class or ability, there is a development of coded categories that persons and groups will be sorted through.[31] As some observers have indicated, if data can be collected on consumers to be used for targeted advertising, then the same systems can be deployed "to identify and isolate groups and persons that may be thought as perpetrators of 'terrorist' acts"[32]. Indeed, this social sorting " ... has become a standard way of discriminating between different persons and groups for the purposes of providing differential treatment".[33] How then, might these same processes be deployed around identifications rooted in gender identity? In the early 1950s, intelligence agencies and law enforcement could use the group affiliations of homosexual men and women to attack pro-gay activist or social networks under the guise of combatting "criminal sexual psychopathy"[34]; clearly then, there is a precedent for such concern.

Borders occupy another, psychically powerful position in national discussions, in addition to their regulatory role as a barrier around the state; borders are both a physical and cognitive barrier around national identity as well as citizenship.[35] The practice of bordering is an act of nation-building, with the border acting as a literal and figurative container of the nation and its citizens.[36] A delineation of the border line determines where Canada begins and ends and therefore serves as a visual and emotional proxy for where *Canadians* exist. Travelers who seek to gain entry to Canada, and ultimately, to become citizens must do so in a manner that is not merely legally valid but narratively satisfying; stories in national media about the transition of refugees from outside petitioners to full-fledged members of the Canadian body-politic emphasize the legal routes taken by individual agents through the regulatory regimes at Canadian borders. These narratives can be contrasted with the greater level of ambivalence displayed by Canadians to the presence of unregulated, illegal border crossings by refugees fleeing the United States in the first year of the Trump Administration[37] and set against a backdrop of relative ambivalence towards refugees in general.[38]

With the move towards biometric bordering, the Canadian state has opened the possibility of an increasingly granular approach to managing the flows of labor, migration, and refugees across its borders. This practice enables the possibility of controlling the flow of specific bodies across borders and through the refugee claims process, such that the state can manage flows not only by category

---

28 (Pötzsch 2015).
29 (Pötzsch 2015).
30 (Pötzsch 2015).
31 (Lyon 2006).
32 (Lyon 2006).
33 (Lyon 2006).
34 (Chernier 2003).
35 (Schwenken et al. 2014).
36 (Wastl-Walter 2011).
37 (Angus Reid Institute 2017).
38 (Denette 2017).

(high-skilled, medium-skilled or low-skilled labor, for example), but by *body type* or even by the *performance* of gender.

For queer migrants, sexuality and gender identity become sites of tension—especially where these identities come into conflict with traditional patterns of gender and sexual identity on display in a state's cultural discourse. For many countries in the Global North—Canada included—specific patterns of gender and sexual identity are woven into the identity of the nation state. As Eithne Luibheid argues, "heterosexuality [is] at once necessary to the state's ability to constitute and imagine itself, [and] simultaneously mark[s] the site of its own instability"[39]. Luibheid points out that exclusion is also an act that delineates what counts as an ideal citizen and further suggests that these arbitrary traits demonstrate the nation's insecurity with regard to its own self-image. Exclusion could be said to protect the traits of those who belong to a nation, therefore strengthening the idea of who rightfully *can* belong.

If identities can affect border crossing mobility, then it is not the geographical border that matters but rather the specific identities in question that are bordered; new technologies can complicate this factor, especially in who can be bordered in the context of their class, race, nationality, gender or sexuality. Queer migrants must often not only contend with the heteronormative culture of their home nation, which might enforce strict adherence to gender binaries or adherence to compulsory heterosexuality, but they also encounter the new nation's own perspectives, which could be similar. The state's border control may in effect serve as an enforcer of preferred patterns of sexuality and gender; both Canada and the US have, at different points in their histories, excluded homosexuals from entering the country, for example, because homosexuality was argued to be a pattern of immoral behavior. In the twenty-first century, biometric border practices may compromise "a critical technology for (re)reproducing national heteronormativity within global and imperial fields"[40] and national heteronormativity.[41]

## 4. Performativity

If one accepts the preposition that national characters—and indeed national identities—are distinctly gendered phenomena, then it must hold true that bordering practices within and around states are similarly gendered. How else, after all, can the gendered national identity of a state be maintained, without consideration given to the gendered identities and performativities of those potential citizens who seek membership? Indeed, the gendered nature of border practices—of bordering—has been recognized for quite some time, from examinations of the narratives of border crossings,[42] to deconstructions of national myths about 'secure borders' or nations as 'safe havens'.[43] The gendering of borders is not isolated to the securitization of nonbinary, queer or trans bodies but of all bodies; young cisgender men—particularly unaccompanied men—are often scrutinized as potential vectors for terrorist activity or simply as potential recruits for extant terrorist networks,[44] which has the practical effect both relaxing *and* hardening border permeability along gender lines; borders are relaxed to accommodate bodies that conform to Western feminine ideals while remaining hardened against those bodies seen as potential vectors of threat, criminality or violence.

Such patterns of gender selectivity are troubling enough, but they occur within a broader context in which the performativities of the women and men being affected are both culturally intelligible and generally accepted within the heteronormative framework of Canadian gender politics. Other cases are not so simple, because while this pattern of gendering can be cause for alarm, it nevertheless occurs in an epistemological space free from gender trouble—where gender performativities can be

---

[39]   (Luibheid 2005).
[40]   (Luibheid 2005).
[41]   (Luibheid 2005).
[42]   (Schimanski 2010).
[43]   (Yuval-Davis et al. 2005).
[44]   (Kingsley 2015; Bielski 2010).

quickly and painlessly mapped to essentialist sex/gender dichotomies. In situations where refugees' gender performativities fail to—or refuse to—map to heteronormative and cisnormative gender dyads, the interpretation of gender (and, therefore, the degree to which the border becomes porous) must be adjudicated by individual customs agents who may find such gender difference unintelligible or even threatening. According to one post from the *Lives in Transition* blog, while some border security or other customs personnel strive to make crossings comfortable, others can instead call an individual's identity into question, by seeing them as either a joke or even a threat.[45]

Despite continued calls for increased sensitivity and understanding on the part of border agents and claims officials for nonbinary, queer, trans, and gender-nonconforming claimants and other travelers, much of the Canadian legal precedent retains an essentialist perspective on the nature of gender and sexual identity, which means that while some groups have found it easier to claim refugee status in recent years, such claims remain problematic for queer and trans claimants.

Although *AG. v. Ward* set the precedent for refugee claims to be made based on fear of persecution due to sexual orientation, it relied on an understanding of sexuality that was rooted in an essentialist understanding of sexual orientation that could potentially serve to exclude sexual minorities other than homosexuals from consideration for refugee status, largely by placing the burden of identifying a claimant's sexual or gender identity up to the processing agents at the border. In effect, it can often come down to individual refugee and immigrations officers to determine what terms like "gay", "transgender", "lesbian", and "queer" mean in an operational context. Without clear—and universal—definitions or understandings, refugee claimants may suddenly find parts of their identities (even, crucially, the very part related to their claims of persecution) being scrutinized for authenticity by agents whose own expertise in such matters may be deeply lacking. Some scholars have argued that the current language of the Act is overly simplistic and that it could endanger potential claimants because the definition of "innate or "unchangeable" carries essentialist definitions regarding the embodiment of sexuality or gender.

The late legal scholar Nicole LaViolette, writing on the language of the *Ward* ruling, noted that:

> ... the decision in *Ward*, while a positive development, inappropriately classifies sexual orientation as an immutable personal characteristic. It suggests lesbians and gay men are deserving of international protection only because they cannot change the personal attribute for which they are persecuted.[46]

The contrapositive to this, of course, is that if a personal attribute is not "immutable" and "unchanging", then they are not deserving of international protections. More practically, what such a ruling means is that refugee claimants whose gender identity or sexual orientation does not fit into conventional, hetero and cisnormative understandings may face discrimination at the border for those characteristics—the same characteristics that are the source of their fears of persecution in the first place.

Put simply, a refugee attempting to cross into Canada might be denied entry for not performing their sexual orientation or gender identity "right" to conform to the border agent's understanding of queerness. Claimants must effectively navigate Canadian cultural understandings of sex and gender and attempt to embody them in order to be recognized as belonging to a marginalized group, even if their own authentic lived experiences differ from Canadian cultural norms.[47] In other words, refugee claimants are incentivized to adopt Western patterns of behavior that reflect Western understandings of gender and sexuality in order to be culturally intelligible to the border agents who serve as gatekeepers to entry. Performances of sexual or gender identity that do not meet the cultural expectations of these gatekeepers may result in claimants feeling forced to "prove" their marginalized position to the satisfaction of border agents or risk deportation or the rejection of their refugee claim. Prospective

---

[45] (Lives in Transition 2015).
[46] (LaViolette 2011).
[47] (Clark 2005).

claimants may not at all identify with this framework or understand it, which presents enormous difficulty in communicating the gravity of their need.

Queer and trans migrants face the challenge of navigating Canadian society's understandings of sexuality or gender that often rely on two prominent narratives. The first of these narratives centers on the performance of the "Respectable Same-Sex Couple" (RSSC), which stands in contrast to the second, that of the sexually promiscuous, "queer" youth, a narrative closely associated with stereotypes of drug use, infidelity, and "morally deviant" practices.[48] As scholar Mariana Valverde argues, the "Respectable Same-Sex Couple" is a new cultural and media phenomenon that shifts the rhetorical representation of queer and trans people as deviants or non-normative people to one that emphasizes the couple's normative performances of marriage, financial independence, and respectability. Indeed, this re-articulation of same-sex partnerships reflects a "normal" heterosexual lifestyle and "proper" gender roles. The performance of such "respectability politics" comes at a cost. [49] In addition, "respectability" is intersectional, as what it means in the Canadian context to be "respectable" (i.e., "normal") is bound to considerations of class and race; a person or couple are respectable insofar as they conform to white, middle-class patterns of behavior.[50] Claimants whose backgrounds place them outside of this normative space may face additional barriers to a successful claim as a result.

However, when we emphasize same-sex partnerships as functionally heteronormative in form and practice and position this performance of queerness as the "correct" pattern of behavior, then a kind of hierarchy of SOGIE performativity is established, and the further any one performance falls from the "respectable" ideal, the less likely it is to be taken seriously or accepted as appropriate by cultural and state apparatuses—including border services.[51]

Opposite the narrative of respectability are the culturally deviant, promiscuous, "alternative" lifestyles that exist in opposition to normative mainstream life in which certain characteristics of sexual minorities are founded on clear and fixed meanings—that there is a clear boundary between heterosexual and homosexual identities that "require a public expression of private of behaviour", for instance.[52] One such example of the cultural stereotype that validates identity is the gay bar, which is often depicted as a space of drugs, desperation, and depravity, an avenue for infectious disease, and other qualities that might betray the RSSC image. Nevertheless, the gay bar is stereotypically a site of community, and attendance is often a signal of one's inclusion in it. These two dominant stereotypes in the Canadian context can inform how queer and trans asylum applicants may be understood, but because these kinds of public expressions of identity are both culturally specific and require a permissive legal context, this narrative poses a challenge to many refugees who attempt to construct a compelling description of their persecution.

When adjudicators assess claims of perceived persecution based on "sexual orientation and gender identity", they may employ their folk knowledge when putting the claims to the test for persecution[53]. While recent memoranda and updated guidelines may facilitate the adoption of basic, shared terminology among adjudicators, it remains unclear how much influence the guidelines will have over individual adjudicators' attitudes, which tend to reflect dominant cultural attitudes and beliefs regarding gender and sexual minorities.

When faced with the knowledge that their sexual or gender identities may be misunderstood, queer migrants are often incentivized to embody Western "coming out" narratives that are "based on a specific cultural and gendered experience of sexuality that may not be more broadly applicable"[54]. These narratives, moreover, may fail to consider why queer migrants would conceal their identity,

---

[48] (Valverde 2006).
[49] (Seidman 2001).
[50] (Lawler 2014).
[51] (Seidman 2001).
[52] (Morgan 2006).
[53] (Lee and Brotman 2011).
[54] (Berg and Millbank 2009).

which could be detrimental to proving one's "credibility" in their claim because if they were not "out" (in the North American understanding of what it means to "come out of the closet"), there is the belief that they did not face violence or persecution.[55]

A second, though equally important, challenge facing SOGIE claimants is that to prove their well-founded fear of persecution by proving their identity as a member of a sexual or gender minority group, they must deliberately—and publicly—perform an identity that they have in many cases spent much of their lives hiding; repression of queer identity is often a survival strategy in states with repressive laws and cultural norms. A lifetime spent hiding one's identity in the face of systemic cissexism and homophobia can leave such claimants with deep feelings of shame or guilt about their identities, which can often translate into a reticence to effectively enact those identities on command.[56] Paradoxically, SOGIE claimants that are afraid to disclose their identity for feelings of shame or fear of discrimination may experience the same form of discrimination that led to the initial hesitance while in midst of the asylum process, especially at the board hearings. When compounded by cultural differences in the ways that SOGIE claimants understand and perform their identities, the asylum process becomes even more complicated, and the risks to the claimants themselves greater. In one example, queer activist and *successful* Canadian refugee claimant Val Kalende returned to their native Uganda after describing incredible difficulty finding work and community in Canada.[57] Their experiences with the Canadian refugee system included being asked to confirm the *stability* of their gender identity by providing letters from ex-partners and friends—some of whom had been murdered in Uganda. In other words, to prove their queerness, Kalende was expected to place their friends and family in Uganda at risk, while also being made to conform to a pattern of gender that included *stability* as an element of its definition, when Kalende's authentic self may have been more fluid. Kalende's experiences are also an important example of how the Canadian refugee system does little to assist SOGIE claimants whose applications have been successful.

As some researchers have pointed out, the ways that SOGIE claimants articulate their gender and sexuality may change over time and, throughout the course of their interactions with immigration agents, those articulations might not always align to Western norms.[58] These fluctuations in identity formation and performance can pose problems when attempting to make claim based on SOGIE, especially under *Ward*'s definition of sexuality as innate and unchanging.

Articulating one's sexual orientation or gender identity poses challenges that add additional difficulty unique to queer asylum claimants that other asylum claimants may not endure. For example, those claiming asylum due to political opinion, nationality, and religion can be independently verified of their belonging to a *particular social group*, whereas SOGIE are held to presenting a highly and individual form of internal identification[59]. These internal identifications are scrutinized by adjudicators such that the claimants' success may be contingent upon their ability to align their sexuality and gender identity along Western, white or dominant embodiments of forming one's sexual identity[60]. SOGIE asylum claimants must prove their internal identity and may be called on to do so by providing objective documentary evidence, such as a current sexual partner, photographs or association with LGBT community groups. Even though adjudicators prefer instances of "objective" evidence to personal narratives, some alarming research indicates that the objective evidence is often "disregarded as self-serving or staged" by individual agents or case workers.[61]

---

[55]   (Lee and Brotman 2011).
[56]   (Berg and Millbank 2009).
[57]   (Jones 2018).
[58]   (Lee and Brotman 2011).
[59]   (Berg and Millbank 2009).
[60]   (Lee and Brotman 2011).
[61]   (Berg and Millbank 2009).

## 5. Summary

Queer and trans refugee experiences at the Canadian border are unstable and fraught with uncertainty, much of which is rooted in a legal landscape that relies on outmoded and empirically dated perspectives about the realities of gender identity and expression. Despite the existence of a legal tradition that ostensibly grants protection and refuge from gender-based persecution and an international reputation as a progressive, refugee-friendly state,[62] Canada's laws governing the regulation and admission of refugees fleeing gender-based persecution remains tied to essentialist beliefs about the innate, hardwired nature of gender. While this sort of reductionist thinking can ease regulatory headaches and provide for more clear-cut legal remedies to pressing issues, it can only do so by erasing all those travelers and refugees whose gender either does not easily map to simple dichotomies or that deny them entirely. In addition, rather than addressing this issue comprehensively through law, Canadian agencies have adopted piecemeal and often short-lived solutions in the form of memoranda of understanding or Chairperson 'guidelines' that have less force and effect (or longevity) than formal law. Queer, trans, and other gender-nonconforming individuals seeking entrance into Canada are therefore faced with the prospect of having multiple interpretations of their gender identity assigned to them by claims agents and border security that may be significantly different from one another.

This obstacle is compounded by the processes of biometric securitization that have become ubiquitous at Canadian points of entry and which rely on the regulation, identification, and differentiation of 'suspicious' bodies from 'normal' ones, leaving many queer and trans claimants on the wrong side of a bordering process that might categorize them as risks simply for performing gender differently.

Like other refugees fleeing gender-based persecution, queer and trans refugee claimants rely on the Canadian state's interpretations of the United Nations Convention on Refugees. Unlike other categories of SOGIE claimants, however, the transitive or fluid nature of trans and queer gender performativities renders any attempt at categorization unstable, and as a result, queer and trans claimants can face additional procedural, bureaucratic, and legal hurdles in their attempts to successfully navigate Canadian refugee systems. To maximize their chances, some trans and queer claimants might feel a strong incentive to alter their gender expression, attempting to force it to conform to cissexual and homonormative narratives of gay or lesbian identity. The result of such attempts is paradox, as queer and trans refugee claimants engage in inauthentic gender performatives as a means of proving their authentic gender-based persecution—forced to live a lie to live their truth.

**Author Contributions:** M.M. was responsible for the initial research and assembling the first draft of this piece. E.H. and H.H. conducted secondary research and provided additional writing and editing.

**Funding:** This research was supported by Borders in Globalization, a Social Sciences and Humanities Research Council of Canada Partnership Grant (Grant no: 895-2012-1022).

**Conflicts of Interest:** The authors declare no conflict of interest.

## References and Note

Agamben, Giorgio. 1998. *Homo Sacer: Sovereign Power and Bare Life*. Stanford: Stanford University Press.

Amoore, Louise. 2006. Biometric borders: Governing mobilities in the war on terror. *Political Geography* 25: 336–51. [CrossRef]

Anderson, Bridget, Matthew J. Gibney, and Emanuela Paoletti. 2011. Citizenship, deportation and the boundaries of belonging. *Citizenship Studies* 15: 547–63. [CrossRef]

---

[62] (Lukacs 2017).

Angus Reid Institute. 2017. Half of Canadians Say Their Country Is 'Too Generous' toward Illegal Border Crossers. *Angus Reid Institute*. September 1. Available online: http://angusreid.org/asylum-seekers-quebec-refugees/ (accessed on 20 December 2017).

Arendt, Hannah. 1973. *The Origins of Totalitarianism*. New York: Houghton Mifflin Harcourt.

Attorney General v. Ward. 1993. 21937 (Supreme Court of Canada, June 30).

Baker, Alicia. 2013. *Securing Bodies: Performances of Security by Transgender Travelers in Canadian Airports and Borders*. Ottawa: Carleton University.

Basaran, Tugba. 2008. Security, law, borders: spaces of exclusion. *International Political Sociology* 2: 339–54. [CrossRef]

Beauchamp, Toby Cason. 2016. When Things Don't Add Up: Transgender Bodies and the Mobile Borders of Biometrics. In *Trans Studies: The Challenge to Hetero/Homo Normativities*. Edited by Yolanda Martinez-San Miguel and Sarah Tobias. New Brunswick: Rutgers University Press, pp. 103–12.

Berg, Laurie, and Jenni Millbank. 2009. Constructing the personal narratives of lesbian, gay, bisexual asylum claimants. *Journal of Refugee Studies* 22: 195. [CrossRef]

Bielski, Zosia. 2010. Burden of Proof. *The Globe and Mail*. April 30. Available online: https://beta.theglobeandmail.com/life/relationships/after-lifetime-of-hiding-gay-refugees-to-canada-expected-to-prove-theiridentity/article34858343/?ref=http://www.theglobeandmail.com& (accessed on 20 September 2017).

Canadian Museum of Immigration at Pier 21. 2017. Immigration Act, 1952. Available online: http://www.pier21.ca/research/immigration-history/immigration-act-1952 (accessed on 9 August 2017).

Chernier, Elise. 2003. The Criminal Sexual Psychopath in Canada: Sex, Psychiatry and the Law at Mid-Century. *Canadian Bulletin of Medical History* 20: 75–101. [CrossRef]

Clark, Anna. 2005. Twilight Moments. *Journal of the History of Sexuality* 14: 139–60. [CrossRef]

Currah, Paisley, and Tara Mulqueen. 2011. Securitizing gender: Identity, biometrics, and transgender bodies at the airport. *Social Research: An International Quarterly* 78: 557–82.

Denette, Nathan. 2017. Sizable Minority Says Canada Is Accepting Too Many Refugees: Poll. *The Globe and Mail*. April 14. Available online: https://www.theglobeandmail.com/news/politics/sizable-minority-says-canada-is-accepting-too-many-refugees-poll/article34087415/ (accessed on 20 December 2017).

Fullerton, Maryellen. 1993. A Comparative Look at Refugee Status Based on Persecution Due to Membership in a Particular Social Group. *Cornell International Law Journal* 26: 505–64.

Government of Canada. 2017. Consolidated Regulations. Available online: http://laws-lois.justice.gc.ca/eng/regulations/SOR-2007-82/page-2.html#docCont (accessed on 20 December 2017).

Hathaway, James C., and Michelle Foster. 1991. *The Law of Refugee Status: Second Edition*. Cambridge: Cambridge University Press.

Immigration and Refugee Board of Canada. 2017. Chairperson's Guideline 9: Proceedings Before the IRB Involving Sexual Orientation and Gender Identity and Expression. Available online: http://www.irb-cisr.gc.ca/Eng/BoaCom/references/pol/GuiDir/Pages/GuideDir09.aspx#a3 (accessed on 15 August 2017).

Jones, Tiffany. 2018. The Year in Queer: Refusal Tweets. *Bent Street: Australian LGBTIQA+ Arts, Writing & Ideas*. Available online: https://bentstreet.net/2018-the-year-in-queer-refusal-tweets-tiffany-jones/ (accessed on 15 June 2019).

Kingsley, Patrick. 2015. Canada's Exclusion of Single Male Refugees May Exacerbate Syrian Conflict. *The Guardian*. November 24. Available online: https://www.theguardian.com/world/2015/nov/24/canada-exclusion-refugees-single-syrian-men-assad-isis (accessed on 20 December 2017).

LaViolette, Nicole. 2011. The immutable refugees: Sexual orientation in Canada (AG) v. Ward. *University of Toronto Faculty of Law Review* 55: 1–41.

Lawler, Steph. 2014. *Identity: Sociological Perspectives*. Cambridge: Polity.

Lee, Edward Ou Jin, and Shari Brotman. 2011. Identity, Refugeeness, Belonging: Experiences of Sexual Minority Refugees in Canada. *Canadian Review of Sociology* 48: 241–74. [CrossRef] [PubMed]

Lives in Transition. 2015. Crossing the Border: Transgender and Non-Binary Experiences. *Lives in Transition: FTM, MTF, Gender Fluid, Non-Gender Conforming, Non-Binary*. August 13. Available online: https://liamrcarter.wordpress.com/2015/08/13/crossing-the-border-transgender-and-non-binary-experiences/ (accessed on 20 December 2017).

Luibheid, Eithne. 2005. Heteronormativity, Responsibility, and Neo-liberal Governance in U.S. Immigration Control. In *Passing Lines: Sexuality and Immigration*. Edited by Brad Epps, Keja Valens and Bill Johnson Gonzalez. Cambridge: Harvard University Press.

Lukacs, Martin. 2017. Justin Trudeau's tweets won't make Canada a refugee haven—But popular pressure can. *The Guardian*. January 31. Available online: https://www.theguardian.com/environment/true-north/2017/jan/30/justin-trudeaus-tweets-wont-make-canada-a-refugee-havenbut-popular-pressure-can (accessed on 21 December 2017).

Lyon, David. 2006. Airport Screening, Surveillance, and Social Sorting: Canadian Responses to 9/11 in Context. *Canadian Journal of Criminology and Criminal Justice* 48: 397–411. [CrossRef]

Morgan, Deborah. 2006. Not Gay Enough for the Government: Racial and Sexual Stereotypes in Sexual Orientation Asylum Cases. *Law and Sexuality Review* 15: 135.

Musalo, Karen. 2003. Beyond Belonging: Challenging the Boundaries of Nationality: Revisiting Social Group and Nexus in Gender Asylum Claims: A Unifying Rationale for Evolving Jurisprudence. *DePaul Law Review* 52: 777–1285.

Parker, Noel, and Nick Vaughan-Williams. 2009. Lines in the sand? Towards an agenda for critical border studies. *Geopolitics* 14: 582–7. [CrossRef]

Pickering, Sharon, and Leanne Weber. 2006. Borders, mobility and technologies of control. In *Borders, Mobility and Technologies of Control*. Edited by Sharon Pickering and Leanne Weber. Dordrecht: Springer, pp. 1–19.

Pickering, Sharon, and Leanne Weber. 2014. New deterrence scripts in Australia's rejuvenated offshore detention regime for asylum seekers. *Law & Social Inquiry* 39: 1006–26.

Pötzsch, Holger. 2015. The emergence of iBorder: bordering bodies, networks, and machines. *Environment and Planning D: Society and Space* 33: 101–18. [CrossRef]

Raj, Althia. 2012. Canada Identity Screening Regulations: Transgendered Community Effectively Banned from Flying. *Huffington Post Canada*. February 1. Available online: http://www.huffingtonpost.ca/2012/01/31/canada-air-travel-transgendered-community_n_1245598.html (accessed on 20 December 2017).

Robinson, Daniel, and David Kimmel. 1994. The Queer Career of Homosexual Security Vetting in Cold War Canada. *The Canadian Historical Review* 75: 319–45. [CrossRef]

Salter, Mark B. 2008. When the exception becomes the rule: Borders, sovereignty, and citizenship. *Citizenship studies* 12: 365–80. [CrossRef]

Salter, Mark B., and Genevieve Piche. 2011. The Securitization of the US-Canada Border in American Political Discourse. *Canadian Journal of Political Science/Revue Canadienne de Science Politique* 44: 929–51. [CrossRef]

Schimanski, Johan. 2010. Reading Gender in Border-Crossing Narratives. In *Gendering Border Studies*. Edited by Jane Aaron, Henrice Altink and Chris Weedon. Cardiff: University of Wales Press, pp. 105–26.

Schwenken, Helen, S. Russ, and Sabine Ruß-Sattar. 2014. *New Border and Citizenship Politics*. London: Springer.

Seidman, Steven. 2001. From identity to queer politics: Shifts in normative heterosexuality and the meaning of citizenship. *Citizenship Studies* 5: 321–8. [CrossRef]

United Nations High Commissioner for Refugees (UNHCR). 2010. Convention and Protocol Relating to the Status of Refugees. *UNHCR: The UN Refugee Agency*. Available online: http://www.unhcr.org/3b66c2aa10 (accessed on 29 December 2017).

*Matter of Acosta*, A-24159781, United States Board of Immigration Appeals, 1 March 1985. Available online: https://www.refworld.org/cases,USA_BIA,3ae6b6b910.html (accessed on 27 June 2019).

Valverde, Mariana. 2006. A new entity in the history of sexuality: The respectable same-sex couple. *Feminist Studies* 32: 155–63. [CrossRef]

Wastl-Walter, Doris. 2011. *The Ashgate Research Companion to Border Studies*. Surrey: Ashgate.

Yuval-Davis, Nira, Floya Anthias, and Eleonore Kofman. 2005. Secure borders and safe haven and the gendered politics of belonging: Beyond social cohesion. *Ethnic and Racial Studies* 28: 513–35. [CrossRef]

