# Peer review of "Performing Borders: Queer and Trans Experiences at the Canadian Border"

_socsci, doi:10.3390/socsci8070201_

Round 1

Reviewer 1 Report

The theoretical article presented is interesting because of the social implications of what is analyzed here. Although some statements are not properly justified by previous research that justifies them, the article as a whole is interesting and I support its publication in its current form.

Author Response

We thank the reviewer for their feedback, and have added additional research to support the claims made in the paper. We have also worked to improve clarity throughout.

Reviewer 2 Report

This is a wonderful and timely paper. I suggest one addition: it would be useful at the start of this paper (or at the section around citation 46) to consider the Val Kalende case of returning to Uganda where Queer friends had been murdered after achieving Canadian asylum through the disruptive queer 'credibility' processes. Queer academics like Stella Nyanzi and Tiffany Jones (2018) have asserted this case as misused by Ugandan and Western media to make claims about Kalende having lied about one or the other of their (queer or straight) identities for community acceptance - these academics argue that a compassionate Queer lens foregrounds 'liveability' above any need for proof of an individual's 'identity stability' in its considerations of the values of identity performativity. Invasive Canadian border politics and the subsequent isolation of some asylum seekers, such that they felt a need to return home; suggest Canada's processes and aftercare 'liveability' needs to be reconsidered.

See 'July' in:

Jones, T. (2018).  The Year in Queer: Refusal Tweets. Bent Street 2(1). pp.158-173. https://bentstreet.net/2018-the-year-in-queer-refusal-tweets-tiffany-jones/

Author Response

We thank the reviewer for their suggestions, and have made some edits to reflect the truly horrific case of Val Kalende (Just below footnote 45). Reviewer 2's suggested source is a welcome and powerful addition to our work. 

Reviewer 3 Report

This is an important and innovative exploration how sexuality and gender identities are shaped at border crossings, focusing on asylum applications and the Canadian case. The author details the history of asylum laws in Canada, how biometric borders enforce raced and gendered norms to produce "threats," and the performativity required of sexual and gender identities that permit crossing for those who successfully perform Western tropes of those identities and erases the identities of all those who do not or cannot for various valid reasons. 

One of the most significant interventions in this piece pertains to how essentialism and immutability operates in asylum cases for sexual and gender identities, and yet it was unclear to me as I was reading where the author stood on how we should understand identity (even while I can guess based on the last section that the author sees these identities as socially constructed and mediated by laws and institutions). To clarify this, I would suggest foreshadowing the ways that norms about identities are imposed on people attempting to cross the Canadian border earlier. This could take place on page 4 of the manuscript and pull from the last section regarding performativity and "folk knowledge" narratives re: these identities. 

The section on biometric borders was strong and could be made even stronger by looking to the work of Toby Beauchamp (2019) and Currah and Mulqueen (2011) on these topics. 

One question I had regarding biometrics was the visability of gender vs the visibility of sexuality. The author states in the conclusion (pg 11) that gender is considered essential and "hard wired." What about sexuality? Again, this could be addressed by the changes suggested for the introduction.

Page 10: The race and class dimensions of respectability politics should be addressed, especially because those who fall out of the respectability narrative seem to be automatically placed in the second "bar goer" narrative. More regarding how norms erase not only sexual and gender identities, but also race and class identities - and thus imperil some who might be more vulnerable - would strengthen the essay. 

Relatedly, and back to essentialism, if biometrics are used to detect fraud, and this is impossible because bodies are not static, and only those with proper documentation can safely pass the border, then this raises the question: for whom do the problems raised in this paper apply? The author alludes to the raced and classed operations, but I think these consequences would be more explicitly articulated in the piece. It is not just essentialism, but something more at play. 

Author Response

We thank the reviewer for their detailed and extremely helpful suggestions. 

We have added some requested "foreshadowing" re: gender identities and expression to page 2 of our paper, to better illustrate what our paper will describe.

We have added an acknowledgment that "respectability" in the Canadian context is not only gendered, but raced and classed as well, and delved somewhat deeper into the ethical and philosophical concerns about pathologizing variance in bodies. What does an "ideal" Canadian immigrant look like in times when algorithms and biometric data pre-screen people for clearance based on body-types?

Reviewer 4 Report

Comments for Review

Overall:

The investigation and premise of this article is strong. It will add very important work to the juncture between queerness and border securitisation which has been often ignored by academia. The paper is well organised and mostly easy to follow.

You need to be clear and define your usage of commonplace queer terms throughout the paper—for many outside the community, your use of transgender, GNC and NB is likely confusing. In order for the work to have a broader impact, consider clarifying in the introduction.  You may also want to clarify the accepted heteronormative gender binary does not apply here. Often when the mainstream considers trans people, they see them as just “switching” to the other gender.

The biometric bordering concept is strong. It could be linked with understandings of both the synopticon and the surveillance of the many by the few with mass data collection, especially at borders. Some of the work that comes out of criminology may be helpful here. I have added some names of authors below.

Keywords:

Nonconforming?? Why? Perhaps gender nonconforming? Would also include “queer”

Introduction:

Consider rewriting the intro-- Jump into the content, instead of what ifs.

Some proofreading for clarity of ideas, shorter sentences, and clunky language. Example, line 21, p 1: For many such crossings (and being crossed) the process remains generally invisible until, suddenly, it is not.

Line 82: Not just unable, but in the case of NB, GNC people, maybe also refuse?

Move between talking about “global north” then North American states. Be clear about setting the context for readers.

History:

Strong overall. Careful about overuse of legal language for lay readers. Also, I would suggest a reading for clarity of ideas within this section as well.  For example:

“Acosta was predicated on an eiusdem generis interpretation of the grounds of persecution found in the refugee convention and as a result, established a legal precedent that linked gender and sexual identity – in all its forms and variations – to immutable, essentialist articulations of gender. Under this legal regime, claimants can only ‘rightly’ be considered refugees if their fears of persecution can be shown to be related to an immutable characteristic of their being; such a predicate would have the effect of diminishing the ability of trans and queer refugee claimants to situate the grounds for their claims in existing legal frameworks, by removing their identities from them entirely.”

Section Three:

Does the border individualise? Or does it in fact tag non-mainstream groups as suspicious. So while an individual may experience the border…it is in fact their difference that marks them for further inspection before and at physical borders.

Missing important lit- Specifically Weber’s work on borders and technology. You bring up carrier sanctions, but don’t name them. There needs to be clarity between state action and private action at state behest. Is it a state border agent making a call on gender or an airline employee?

Consider giving a nod to the citizenship lit when discussing national identity. The idea of conforming and national identity complements this discourse.

Perfomativity:

Consider Cochrane’s work on motherhood and harms in re RAS mothers. Making a broad brush claim about ease of mothers is not bounded in the lit. 

P9 “its advocates establish a hierarchy of SOGIE performativity” Correct for clarity.

P10 “since it is often difficult for non-Canadian refugees to embody the “lifestyle”” Explain why.

Summary

“Canadian agencies are forced to adopt piecemeal and often short-lived solutions in the form of memoranda of understanding or Chairperson ‘guidelines’ that have less force and effect (or longevity) as formal law.” This needs to be expanded/explained.

Author Response

We thank the reviewer for their detailed and supportive suggestions. We have worked to address them within the existing framework of the paper. 

we have re-written our introduction to get to the case at hand more effectively. We feel this has made our paper stronger overall. 

Updated our keywords

We have updated our discussion of borders and screening to include important work by Pickering and Weber, among others. 

We have made substantial edits throughout the paper to improve clarity.

Round 2

Reviewer 3 Report

The revised version of this paper is very well executed. I have no further suggestions for improvements or revisions.